# Construction of the Guide Star Catalog for Double Fine Guidance Sensors Based on SSBK Clustering

**DOI:** 10.3390/s22134996

**Published:** 2022-07-02

**Authors:** Yuanyu Yang, Dayi Yin, Quan Zhang, Zhiming Li

**Affiliations:** 1Shanghai Institute of Technical Physics, Chinese Academy of Sciences, Shanghai 200083, China; yangyuanyu@mail.sitp.ac.cn (Y.Y.); zhangquan@mail.sitp.ac.cn (Q.Z.); lizhm@shanghaitech.edu.cn (Z.L.); 2School of Electronic, Electrical and Communication Engineering, University of Chinese Academy of Sciences, Beijing 100049, China; 3Key Laboratory of Infrared System Detection and Imaging, Chinese Academy of Sciences, Shanghai 200083, China; 4School of Information Science and Technology, ShanghaiTech University, Shanghai 201210, China

**Keywords:** fine guidance sensor, guide star catalog, spherical spiral reference point, binary K-means clustering

## Abstract

In the Chinese Survey Space Telescope (CSST), the Fine Guidance Sensor (FGS) is required to provide high-precision attitude information of the space telescope. The fine star guide catalog is an essential part of the FGS. It is not only the basis for star identification and attitude determination but also the key to determining the absolute attitude of the space telescope. However, the capacity and uniformity of the fine guide star catalog will affect the performance of the FGS. To build a guide star catalog with uniform distribution of guide stars and catalog capacity that is as small as possible, and to effectively improve the speed of star identification and the accuracy of attitude determination, the spherical spiral binary K-means clustering algorithm (SSBK) is proposed. Based on the selection criteria, firstly, the spherical spiral reference point method is used for global uniform division, and then, the K-means clustering algorithm in machine learning is introduced to divide the stars into several disjoint subsets through the use of angular distance and dichotomy so that the guide stars are uniformly distributed. We assume that the field of view (FOV) is 0.2° × 0.2°, the magnitude range is 9∼15 mag, and the threshold for the number of stars (NOS) in the FOV is 9. The simulation shows that compared with the magnitude filtering method (MFM) and the spherical spiral reference point brightness optimization algorithm (SSRP), the guide star catalog based on the SSBK algorithm has the lowest standard deviation of the NOS in the FOV, and the probability of 5∼15 stars is the highest (over 99.4%), which can ensure a higher identification probability and attitude determination accuracy.

## 1. Introduction

Large-scale sky surveys and deep space observations are key directions of international astronomical research, which will greatly promote the study of dark matter, dark energy, the origin and evolution of objects, and other fundamental issues in astronomy and physics [1]. The Chinese Survey Space Telescope (CSST) is planned to be launched at the end of 2023; it is expected to provide humankind with fresh knowledge about distant galaxies, mysterious dark matter, dark energy, and the past and future evolutions of the universe [2,3]. Space telescopes have the advantages of being free from atmospheric interference, having a wide range of observable bands, low background noise, and high angular resolution, which provide favorable conditions for astronomical observation. However, since space telescopes are based on spacecraft, the attitude changes of spacecraft, orbital precession, slight vibration of moving parts, and even the movement of astronauts can affect space exploration and even fail to capture observation targets [4]. At present, space telescopes with high precision and high stability all contain the fine image stabilization system (FISS), which is mainly used to suppress platform jitter and achieve the expected detection accuracy of space telescopes. The principle of the line of sight (LOS) jitter detection in the FISS is to configure the fine guidance sensor (FGS) at the edge of the field of view (FOV) of the telescope to image the stars alone and to calculate the relative attitude change of the LOS of vision by extracting the star position deviation between different frames based on the principle that the star celestial sphere position is unchanged in a short time [5,6].

FGS is one of the main components of the FISS and is an important guarantee module for high-precision astronomical observation. It mainly performs the acquisition, recognition, and tracking of stars in the FOV and provides data for attitude determination, fine pointing, and attitude stability [7,8,9]. The double fine guidance sensor system architecture is shown in Figure 1. In FGS, the fine guide star catalog is the basis of star identification and attitude determination. A reasonable construction of a guide star catalog can improve the speed of star identification and the accuracy of attitude determination.

The Hubble Space Telescope (HST) was launched in 1990, and it contains three sets of FGSs: two for pointing and locking onto objects and one for measuring the position of stars [10]. The Guide Star Catalog (GSC) used by HST contains about $1.9\times 10^{7}$ stars and other objects in the magnitude range of 6∼15. The selection criteria of the guide star are determined by the size of the FOV of the fine guidance sensors, the size of the aperture of the scientific instrument, the accuracy of the inertial system provided by the gyroscope, the expected observation rate, and the orbit constraints of the spacecraft [11]. The second generation of GSC is already employed at GEMINI, VLT, the Large Sky Area Multi-Object Fiber Spectroscopic Telescope (LAMOST), and the Global Astrometric Interferometer for Astrophysics (GAIA) [12].

The James Webb Space Telescope (JWST), the successor to the HST, was launched in 2021. It is mainly used to observe and study the formation of the first generation of galaxies after the Big Bang and the formation of extrasolar planetary systems [13,14]. The selection of guide stars and field identification will be performed by using the Guide Star Catalog 2 (GSC-2) [15]. The working model of the FGS mainly includes the identification process, capture process, and precision guide star process. It uses the method of electronics to search and capture guide stars. The FGS contains two independent 2 k × 2 k infrared focal plane arrays covering the wavelength range from 0.6 to 5 µm. Once in fine guiding, the FGS will provide continuous pointing information to a precision of ∼5 milli-arc seconds (mas) at an update rate of 16 Hz for J ≃ 18.5 magnitude stars [16].

The Spitzer Space Telescope (SST) is an infrared space telescope launched by NASA in 2003. Its guide star catalog contains 196,087 guide stars. According to the guide star selection requirements, it is necessary to create a guide star catalog with a positioning error of no more than 0.05 arcsec over the entire mission period, a magnitude of 7 to 10 in the V-band, and a strict no-nearest neighbor requirement [17]. The guide star catalog was developed in stages. The candidate guide stars were initially selected using Tycho, Tycho-2, Hipparcos, and Tycho Double Star catalogs, and then, the interference effects of neighboring objects were calculated using 2MASS PSC, USNO A2.0, 2MASS XSC, and PGC star catalogs. Finally, the digitized sky survey (DSS) is used to determine the sky background disturbance effect of each candidate guide star [18].

The Euclid Space Telescope (Euclid) is scheduled to launched in 2022. It uses the Gaia catalog as a source catalog. Euclid’s star catalog is divided into the On-Board Star Catalog (OBC) and ground Input Star Catalog (ISC). The ISC stored on the ground is representative of the entire sky, including star data information provided by Osservatorio Astrofisico di Torino (OATO). OBC is a subset of the complete star catalog, including part of ISC. It is prepared by the On-Ground Algorithm (OGA) according to the Euclidean observation plan and extracts specific parts of the ISC and sends it to the spacecraft for FGS operations in the planned sky survey area. The purpose of FGS OBC generation is to provide enough targets to reliably perform attitude recognition and optimize attitude accuracy [19,20,21,22].

The research on the construction of the guide star catalog is more in the direction of the star sensor. The most traditional method is the visual magnitude threshold filtering method (Visual Magnitude Threshold, VMT). For example, Jiang et al. [23] select stars whose value is less than or equal to a certain threshold in the source catalog as guide stars to form the guide star catalog. The guide stars in the guide star catalog constructed by this method can be well-matched with the observed stars, but the uniformity of the star catalog is poor, which easily leads to the redundant guide stars in the dense region and the holes in the sparse region. Therefore, many scholars have proposed the construction method of the star catalog based on the uniform distribution of guide stars.

Li et al. [24] proposed a method for generating a guide star catalog for a star sensor. The probability model was obtained by quantifying brightness and uniformity criteria; in addition, the relationship between the FOV, brightness accuracy, the average number of stars (NOS) in the FOV, and the distribution of guide stars have been established. When the brightness noise of the star sensor is 0.2, the FOV is 16°, and the average NOS in the FOV is 16, the star guide catalog is more evenly distributed, and the standard deviation is 2.39, which is 76% lower than the source catalog. Somayehee et al. [25] proposed a method to make the star catalog uniform using the uniform distribution of points on the celestial sphere and Delaunay’s triangulation method. For a non-uniform star catalog, the FOV must be at least 12.5°, and the probability of observing four or more stars can be greater than 95%. For a uniform star catalog, the FOV needs to be more than 13 degrees. Somayehee et al. [26] proposed a weighted k-means clustering based on the geodesic criterion. According to the algorithm, for a 12° FOV, the probability of observing more than 20 stars in the non-uniform star catalog is about 11.47%, and the probability of observing more than 20 stars in the uniform star catalog is about 0.02%; Zhang et al. [27] proposed a method for generating a high-precision guide star catalog based on a machine learning classification algorithm. The average number of guide stars in the FOV generated by the KNN method is 44.73, and the minimum number of guide stars is 15, which is superior to other classification methods in storage, uniformity, and integrity. Wu [28] proposed a guide star catalog generation algorithm for interstellar star identification, studied the update period of the star catalog, and gave a block update method based on distance distribution.

The navigation systems based on the above methods have a relatively large FOV, low magnitude, and detection accuracy requirements are not particularly high. As the FGS has relatively high requirements on the identification rate and the accuracy of the attitude determination, it is necessary to study the construction of the high-precision fine guide star catalog. According to requirements for the NOS of FOV, the storage capacity, uniformity, identification rate, and the accuracy of attitude determination, we propose a method for constructing a fine guide star catalog based on the spherical spiral binary K-means clustering algorithm (SSBK). Firstly, we select the Gaia Data Release 2 (GaiaDR2) as the source catalog and preprocess it including star pre-screening, magnitude correction, position correction, and double stars elimination. Secondly, according to the selection criteria, we propose a guide stars selection method. Then, we generate the guide star catalog and present the experiment results. Finally, a test considering 10,000 random boresight directions of the entire celestial sphere was conducted to evaluate the guide star catalog. The guide stars of the guide star catalog constructed by it are more evenly distributed, with a smaller standard deviation of the NOS, and the NOS in the FOV is concentrated in 5∼15.

## 2. Preprocessing of the Source Catalog

A star catalog is a database table, which usually records information about object parameters, including position (right ascension, declination), magnitude, proper motion, parallax, type, spectral type, etc. Commonly used star catalog information is shown in Table 1.

Due to the requirement of the accuracy of the attitude determination, the GaiaDR2 is selected as the source catalog. It contains the positions and G magnitudes of 1,692,919,135 objects; the G-band range is 330 to 1050 nm. The GaiaDR2 passbands are shown in Figure 2; the colored lines in the figure show the revised passbands for G, ${G_{B}}_{P}$, and ${G_{R}}_{P}$. Table 1 shows that for objects brighter than 15 mag (magnitude), the accuracy of position and parallax is 0.02∼0.04 mas, and the accuracy of proper motion is 0.07 mas/year. For objects equal to 17 mag, the accuracy of position and parallax is 0.1 mas, and the accuracy of proper motion is 0.2 mas/year. For objects equal to 20 mag, the accuracy of position and parallax is 0.7 mas, and the accuracy of proper motion is 1.2 mas/year.

The preprocessing mainly includes star pre-screening, magnitude correction, position correction, and double stars elimination.

### 2.1. Star Pre-Screening and Magnitude Correction

Not all objects in the source catalog can be selected as guide stars, so star pre-screening is required. Because binary stars are very close to each other and revolve around each other, they will affect attitude determination. A variable star is a star whose brightness is unstable and changes frequently, which is not conducive to detection and recognition. The instability of high proper motion stars and non-stellar objects will affect the accuracy of star identification, so the above objects need to be removed [4].

Magnitude is a measure of a star’s brightness, and it is inversely proportional to its brightness. The greater the magnitude, the dimmer the star. As the temperature of the star is different, its spectral range is different, and the spectral response of different image sensors is also different, so the magnitude value is related to the instrument [29]. GaiaDR2 magnitude values are based on the G-band, which ranges from 330 to 1050 nm. In practical application, the spectral response of the image sensor is different from that of GaiaDR2, so magnitude correction is needed.

Under the requirement of a certain signal-to-noise ratio (SNR), the observable effective magnitude range of the photoelectric system is limited. The highest observable magnitude depends on the noise level of the photoelectric system, while the lowest observable magnitude depends on the dynamic range of the image sensor [30]. Therefore, it is necessary to select the magnitude range of stars in the source catalog. After the above processing, we can obtain the candidate star catalog.

### 2.2. Position Correction

GaiaDR2 contains the location and proper motion data of objects at the time of J2015.5, and the primary coordinate system is the Barycentric Celestial Reference System (BCRS). Its axis is aligned with the International Celestial Reference System (ICRS) [31].

The calculation of the apparent position of a star needs to convert the position of the star in the star catalog from the epoch time to the observation time. The position at the epoch time is calculated by proper motion to J2000.0, and then through proper motion correction, parallax transformation, aberration and light deflection correction, and precession–nutation transformation [32], the apparent position of the star at the time of observation is calculated. The process is shown in Figure 3. α, δ, *pr*, *pd*, *px*, and *rv* represent the star’s right ascension, declination, proper motion of right ascension, proper motion of declination, parallax, and radial velocity, respectively [33].

### 2.3. Double Stars Elimination

For the two stars that are very close to each other in the boresight direction and cannot be distinguished in the focal plane, they are called double stars, which will interfere with the star identification and affect the recognition of other stars. Because the size of the imaging star point is related to the point spread function (PSF) of the optical system, to improve the positioning accuracy of the star point position, the star point is usually expanded to 3 pixel × 3 pixel or 5 pixel × 5 pixel through defocusing. Figure 4 is a schematic diagram of double stars when the radius of the point spread function is 1 pixel. Suppose the star point size is *N* pixel × *N* pixel, the FOV size is *FOVh° × FOVw°* (height × width), and the detector size is *PixV × PixH*; then, the angular distance threshold of the double stars is:(1)$$\theta_{\mathrm{th}}=N_{p}\times \frac{FOVw}{PixH}$$
where $N_{p}$ is the pixel number of defocus diffusion.

Therefore, the double stars whose angular distance between stars in the FOV is less than $\theta_{\mathrm{th}}$ should be removed.

## 3. Guide Star Selection

A guide star catalog with a small number of guide stars, uniform distribution, and high integrity is beneficial to improve the efficiency of star identification and star tracking. Therefore, it is necessary to divide the stars in the whole celestial sphere and select the guide stars in the region. The selection of guide stars in the region is related to the algorithm of star identification and attitude determination. Too many stars in identification will greatly affect the time and efficiency. Too few stars may cause identification failure, the NOS in the range of 5∼15 stars can be guaranteed a high identification probability, at least three stars are required in the FOV of the detector in the attitude determination, and nine stars can improve the attitude accuracy [20], and in the actual identification and matching, if the brightness of the observed stars is higher, the accuracy of centroid extraction will be higher. Therefore, based on the selection criteria of satisfying the above requirements and making the size of the star catalog as small as possible and evenly distributed as possible, a guide star selection algorithm based on spherical spiral binary K-means clustering (SSBK) is proposed to construct a fine guide star catalog.

The flow chart of the SSBK is shown in Figure 5. It is mainly composed of two parts: one is the selection of the spherical spiral reference point, and the other is the binary K-means clustering algorithm.

### 3.1. Selection of the Spherical Spiral Reference Point

To improve the uniformity of guide stars distribution, the whole celestial sphere should be evenly divided into regions close to the size of the observation field. Bauer et al. [34] proposed a method for selecting spherical spiral points in which the spacing between spiral circles is almost the same as that between points along the spiral, making the spiral point set very uniform. The calculation of n spiral points on a unit sphere is as follows:(2)$$\left\{\begin{array}{c} L_{n}=\sqrt{\pi n}\\ z_{k}=1-\frac{2k+1}{n}\\ \phi_{k}=\arccos\left(z_{k}\right)\\ \theta_{k}=L_{n}\phi_{k}\\ x_{k}=\sin\left(\phi_{k}\right)\cos\left(\phi_{k}\right)\\ y_{k}=\sin\left(\phi_{k}\right)\sin\left(\phi_{k}\right) \end{array}\right.$$
where *k* = 0, 1, ⋯, *n*−1.

Since the subsequent calculation is in spherical coordinates, the spiral point needs to be converted from rectangular coordinates to spherical coordinates:(3)$$\left\{\begin{array}{c} \alpha_{k}=\arctan\frac{y_{k}}{x_{k}}\\ \delta_{k}=\arcsin\left(z_{k}\right) \end{array}\right.$$

The value of n is determined by the ratio between the FOV of the whole celestial sphere and the FOV of the FGS. It is known that the FOV of the FGS is *FOVh × FOVw* deg${}^{2}$; then, its corresponding spherical area is:(4)$$\Omega_{\mathrm{FGS}}=\frac{\pi }{180}\times \frac{\pi }{180}\times FOVh\times FOVw$$

The whole celestial sphere Ω=4π, so the ratio of the whole celestial FOV to the FOV of the fine guidance sensor is:(5)$$\begin{array}{cc} n=\frac{\Omega }{\Omega_{\mathrm{FGS}}} & =\frac{4\pi }{\frac{\pi }{180}\times \frac{\pi }{180}\times FOVh\times FOVw}\\ \; & =\frac{129600}{\pi \times FOVh}\times FOVw \end{array}$$

### 3.2. Calculation of the Range of Right Ascension and Declination Corresponding to the FOV

As the boresight direction changes, the range of right ascension and declination corresponding to the FOV will also be different, so it is necessary to calculate the range of right ascension and declination corresponding to the FOV with the spherical spiral point as the center of the boresight. Figure 6 shows a schematic diagram of the FOV corresponding to the right ascension and declination range. If the FOV center (*ra*0, *dec*0) is known, A(ra0−φ,dec0+θ), B(ra0+φ,dec0+θ), C(ra0−φ,dec0−θ), and D(ra0+φ,dec0−θ) can be obtained.

According to the vector cosine formula, the relationship between the FOV and the range of right ascension and declination can be obtained as shown in Equation (Equation 6).
(6)$$\left\{\begin{array}{c} \cos(FOVh)=\cos(2\theta )\\ \cos\left(FOVw\right)=\cos^{2}\left(dec0+\theta \right)\left(\cos\left(2\varphi \right)-1\right)+1 \end{array}\right.$$
where *FOVh* is the height of the FOV, and *FOVw* is the width of the FOV.

Therefore, the calculation formula for the range of right ascension and declination can be obtained:(7)$$\left\{\begin{array}{c} \theta =FOVh/2\\ \varphi =\arccos(\frac{\cos(FOVw)-1}{\cos^{2}\left(dec0+\theta \right)}+1)/2 \end{array}\right.$$

It can be seen from Equation (Equation 7) that in the northern celestial sphere, with the increase of declination, the range of declination corresponding to the FOV remains unchanged, while the range of right ascension changes with the change of declination to which the boresight direction points. Figure 7 shows the curve of the range of right ascension corresponding to the FOV as declination changes (northern celestial sphere).

### 3.3. Binary K-Means Clustering Algorithm

In machine learning, clustering is an unsupervised learning method, which is used to find the internal distribution structure of the data and divide the unlabeled input sample into several disjoint subsets, and each subset is called a “cluster”.

The centroid clustering algorithm is a more suitable method for the problem of the uniformity of the star catalog [26]. Our data are the position of the star in the celestial coordinate system, which is normalized, and the data in the field of view are scattered. The K-means algorithm has the advantage that it produces tighter clusters when dealing with normalized data.

The K-means clustering algorithm is a centroid clustering algorithm, which calculates the similarity of each cluster through Euclidean distance. It can guarantee better scalability when processing large data sets, but it sometimes ends with a local optimum, is prone to empty clusters, and is sensitive to the initial cluster center. To reduce the impact of the above problems, a binary K-means clustering algorithm based on angular distance is proposed.

The binary K-means clustering algorithm mainly uses the angular distance to calculate the similarity of all the input stars. It is similar to a decision tree, which judges whether to divide this cluster into two clusters by the total error after dichotomy. By minimizing the error function, all the input star data are divided into N clusters, and then, the brightest star in the N clusters is selected. The process steps are as follows:All stars in the FOV are called out by calculating the spherical spiral reference point and the right ascension and declination range of the FOV;Set the initial cluster i = 1, and the initial centroid is the average position of all data;Binary K-means clustering is performed on the i clusters, and each cluster is divided into two clusters. Then, calculate the total error after binary, and divide the cluster with the smallest total error into two. If the number of stars in the cluster is less than 2, the cluster will not be divided anymore;i = i + 1, repeat the third step and fourth step until i is less than the threshold N;Finally, the brightest stars in the N clusters are selected as guide stars.

The algorithm is shown in Figure 8. Subgraph 1 represents the stars contained in FOV; all stars are a cluster, which is divided into two disjoint clusters by binary K-means clustering. As shown in subgraph 2, the blue circle represents the first cluster, and the green circle represents the second cluster. Then, divide the two clusters in subgraph 2 into two, respectively, and calculate the SSE, as shown in subgraph 3. The SSE after the cluster in the green circle is divided into two is 50, and the SSE after the cluster in the blue circle is divided into two is 70. Therefore, the green circle with the smallest SSE is chosen to be divided into two clusters, and we obtain three clusters, as shown in subgraph 4. Then, we divide the three clusters in subgraph 4 into two, respectively, and calculate the SSE. As shown in subgraph 5, the SSE after the cluster in the green circle is divided into two is 40, and the SSE after the cluster in the red circle is divided into two is 30. The SSE after the cluster in the blue circle is divided into two is 20, so we choose to divide the cluster in the blue circle into two and obtain four clusters, as shown in subgraph 6. Finally, N clusters can be obtained by binary K-means clustering.

The total error in the algorithm is calculated by Equation (Equation 8), which represents the sum of squares of the distance between the sample point and the centroid $C_{i}$ of cluster i, $dist(x,C_{i})$ represents the angular distance and the ${d_{P}}_{M}$ represents the angular distance between P and M in Figure 9.
(8)$$SSE=\sum_{i=1}^{N}\sum_{x=C_{i}}dist{\left(x,C_{i}\right)}^{2}$$

## 4. Results of Guide Star Catalog Construction

In this paper, the FOV of the FGS is 0.2° × 0.2°, the size of the detector is 2048 × 2048, the waveband range is 400 to 800 nm, and the magnitude selection range is 9∼15 mag. The position was corrected using Standards of Fundamental Astronomy (SOFA) software package provided by the International Astronomical Union (IAU). We fixed the geocentric apparent position to 2022-06-01 00:00:00(UTC). The FOV of the FGS is 0.04 deg${}^{2}$, and the average FOV of the celestial sphere is 1,031,324. To make the FOV overlap, the spherical spiral reference point of the celestial sphere is set as 1,050,000, and the threshold value of the number of guide stars in the FOV is 9.

Figure 10 and Figure 11 show the star distribution and density of the candidate star catalog (CSC) after preprocessing and the guide star catalog based on the SSBK algorithm. The total NOS in the CSC is 104,350,738 and that in the guide star catalog is 9,298,862 which is decreased by 91.09% compared to the CSC.

Figure 10a and Figure 11a show the distribution density of the CSC and guide star catalog in the celestial sphere, in which the color in the color bar represents the magnitude of density, and the value above the color bar represents the NOS in this region. The celestial sphere is divided into 1800 × 900 parts in the direction of right ascension and declination, and the color of each part represents the NOS in this region. It is shown that the distribution of the guide star catalog is more uniform than that of the CSC, and the NOS in the region in Figure 11a is less. By comparing the two images, we can conclude that both have high integrity, but the star guide catalog has smaller data capacity and better redundancy.

Figure 10b,c and Figure 11b,c show the statistics of the NOS distributed in the direction of declination and right ascension. The direction of declination is divided into 900 parts in Figure 10b and Figure 11b, and that of right ascension is divided into 1800 parts in Figure 10c and Figure 11c; the NOS in each part is counted. It can be seen from these figures that stars in the declination direction in CSC are mainly concentrated in about −20° to −60°, and stars in the right ascension direction are mainly concentrated in about 250° to 300°. In the guide star catalog, stars in the direction of declination are mainly concentrated at ±40°, and stars in the direction of right ascension are very even; the NOS of it is greatly reduced compared with CSC.

Figure 10d and Figure 11d exhibit the three-dimensional distribution density of the CSC and guide star catalog in the celestial sphere. Through the comparison of these two figures, it is evident that the data capacity of the guide star catalog constructed based on the SSBK algorithm is greatly reduced, and the guide star distribution is more uniform.

## 5. Evaluation of the Guide Star Catalog

The performance test of the guide star catalog consists of two parts. One is to use the Monte Carlo method [35] to simulate the boresight direction and count the NOS in the FOV. The other part is to calculate uniformity.

The Monte Carlo method is mainly used to randomly generate 10,000 boresight directions (α,δ) in the celestial sphere and count the NOS in each FOV.

The calculation of uniformity mainly includes local uniformity and global uniformity [36,37]. Local uniformity is the measurement of the change in the NOS in a small region of the constant area as it moves from one position on the sphere to another. The standard deviation of NOS is defined as:(9)$${S_{t}}_{d}=\sqrt{\frac{\sum_{i=1}^{N}{\left(N_{i}-\bar{N}\right)}^{2}}{K}}$$

The selection method is as follows: K is 1296, and the initial boresight direction is [−89°, 1°]. The internal cycle is increased by 10° each time along the right ascension until 351°; the external cycle increases by 5° along with declination until 86°.

Global uniformity refers to the measurement of the distribution uniformity of stars throughout the celestial sphere. Suppose the whole celestial sphere has $N_{c}$ stars, and the direction vectors of each star form an $N_{c}$ by three-dimensional matrix A. Then, the global uniformity can be expressed as:(10)$$\Phi =\sum_{i=1}^{3}\delta_{i}\ln\left(3\delta_{i}\right)$$
where Φ is the measurement of the deviation between the distribution of eigenvalues and uniformity. Φ=0 is the ideal state of completely uniform distribution, so the closer the Φ is to 0, the more uniform the distribution is. $\delta_{i}$ is the eigenvalue of the matrix $\left(1/N\right)A^{T}A$, $\left(1/N\right)A^{T}A$ is a 3 × 3 dimensional matrix, the three eigenvalues are non-negative real numbers, and the sum is 1.

Figure 12 shows the position distribution diagram of FGS. The NOS in the FOV and the probability of the occurrence of different NOS are calculated by randomly generating 10,000 boresight directions of the telescope (α and δ).

Table 2, Table 3 and Table 4 shows the statistical results of single FOV and double FOVs. Single FOV is the 0.2° × 0.2° FOV centered on the telescope’s boresight, and double FOVs refer to the FOV of FGS1 and FGS2. In these tables, the candidate star catalog and the three algorithms are compared in terms of guide star catalog capacity, NOS in the FOV, the occurrence probability of stars in the FOV, and uniformity of the star catalog.

Table 2 shows the statistical results of a single FOV. As can be seen from it, in the CSC, there are 101.89 stars on average in the FOV, and the standard deviation of the NOS is 179.10. The probability of 5∼15 stars in the FOV is 9.66%, and the probability of more than 15 is 90.25%. The NOS in FOV is relatively large, and the distribution of stars is not uniform. By comparing the statistical results of the star guide catalog constructed by the magnitude filtering method (MFM), spherical spiral reference point brightness optimization algorithm (SSRP), and the proposed algorithm (SSBK), it can be seen that the three methods greatly reduce the capacity of the star catalog, and the average NOS in the FOV is nearly 9. However, compared with MFM and SSRP, the SSBK-based star number has the lowest standard deviation, and the probability of 5∼15 stars in the FOV is 99.39%, showing better local uniformity.

Table 3 and Table 4 show the statistical results of double FOVs. In FGS1, the probability of 5∼15 stars in the FOV is 99.41%, the probability of 3 or more stars is 99.93%, and the probability of more than 15 stars is 0.07%. In FGS2, the probability of 5∼15 stars in the FOV is 99.59%, the probability of three or more stars is 99.94%, and the probability of more than 15 stars is 0.04%. Through the comparison of the three algorithms, it can be concluded that the guide star catalog based on the SSBK algorithm has the lowest standard deviation of the NOS in the FOV, with an average NOS is 9, and the probability of 5∼15 stars is the highest.

## 6. Conclusions

According to the guide star selection criteria, this paper proposes a guide star catalog construction algorithm based on spherical spiral binary K-means clustering. It starts from both global and local aspects. First, the FOV of the whole celestial sphere is divided evenly by the spherical spiral reference point method from the global aspect, and then guide stars in each FOV are selected through binary K-means clustering from the local aspect. The construction results show that when the FOV is 0.2° × 0.2°, the magnitude range is 9∼15 mag, and the threshold for the NOS in the FOV is 9, the total NOS in the constructed guide star catalog is 9,298,862, which is decreased by 91.09% compared to the CSC. We conducted a the simulation analysis of the performance of the guide star catalog. In FGS1, the standard deviation is 1.67, the average NOS in FOV is 9.0, the probability of 5∼15 stars in the FOV is 99.41%. In FGS2, the standard deviation is 1.65, the average NOS in FOV is 9.03, and the probability of 5∼15 stars in the FOV is 99.59%. Compared with MFM and SSRP, the guide star catalog based on the SSBK has the lowest standard deviation of the NOS in the FOV, with an average NOS is 9, and the probability of 5∼15 stars is the highest, showing better uniformity. This shows that the constructed guide star catalog fully satisfies the guide star selection criteria, and it lays a solid theoretical research foundation for the future use of the high-precision FGS on-orbit star catalog. However, the large number of spherical spiral reference points involved in the calculation and the iterative method adopted in the clustering process leads to a long generation time of the guide star catalog.

FGS is used to output high-precision attitude, so we should combine the guide star catalog with a star identification algorithm and attitude determination algorithm to calculate attitude in real time in the future.

## Figures and Tables

**Figure 1 sensors-22-04996-f001:**
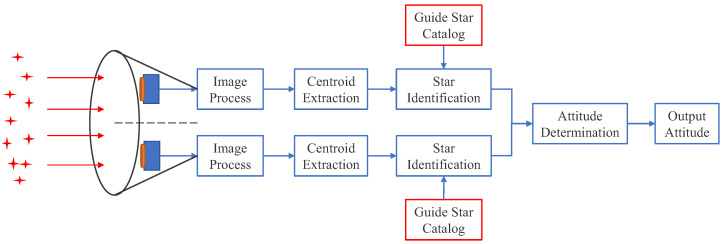
Double fine guidance sensors system architecture.

**Figure 2 sensors-22-04996-f002:**
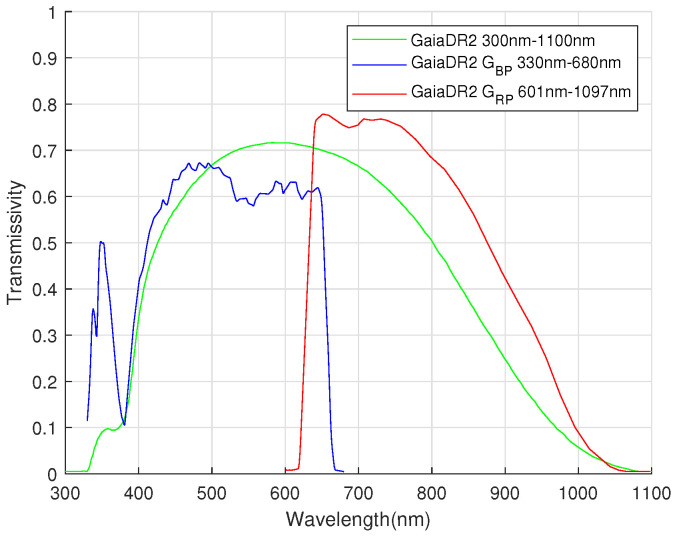
GaiaDR2 passbands (green: G; blue: ${G_{B}}_{P}$; red: ${G_{R}}_{P}$).

**Figure 3 sensors-22-04996-f003:**
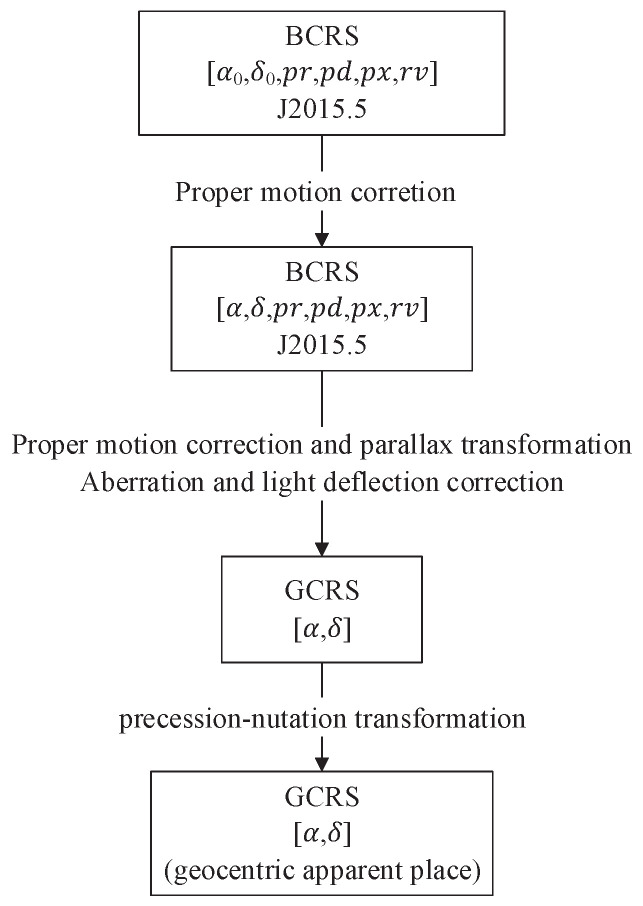
The calculation process of the star’s apparent position.

**Figure 4 sensors-22-04996-f004:**
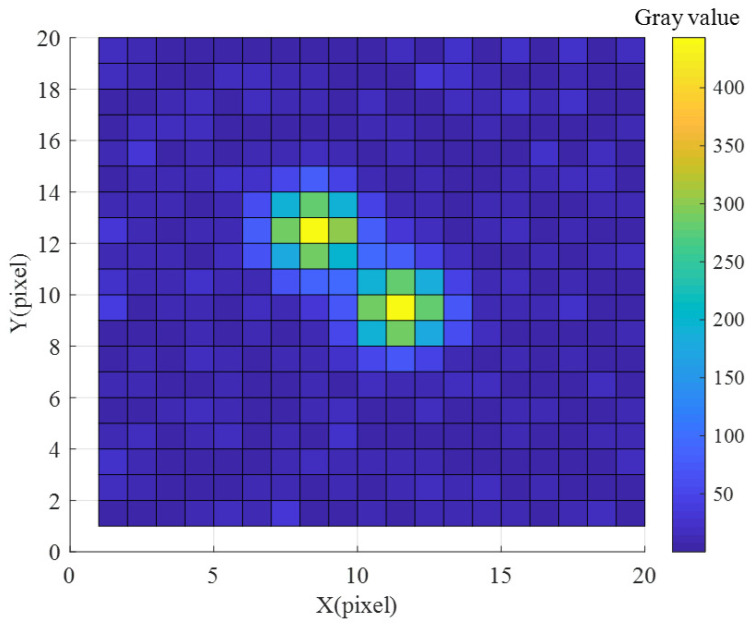
Schematic diagram of double stars.

**Figure 5 sensors-22-04996-f005:**
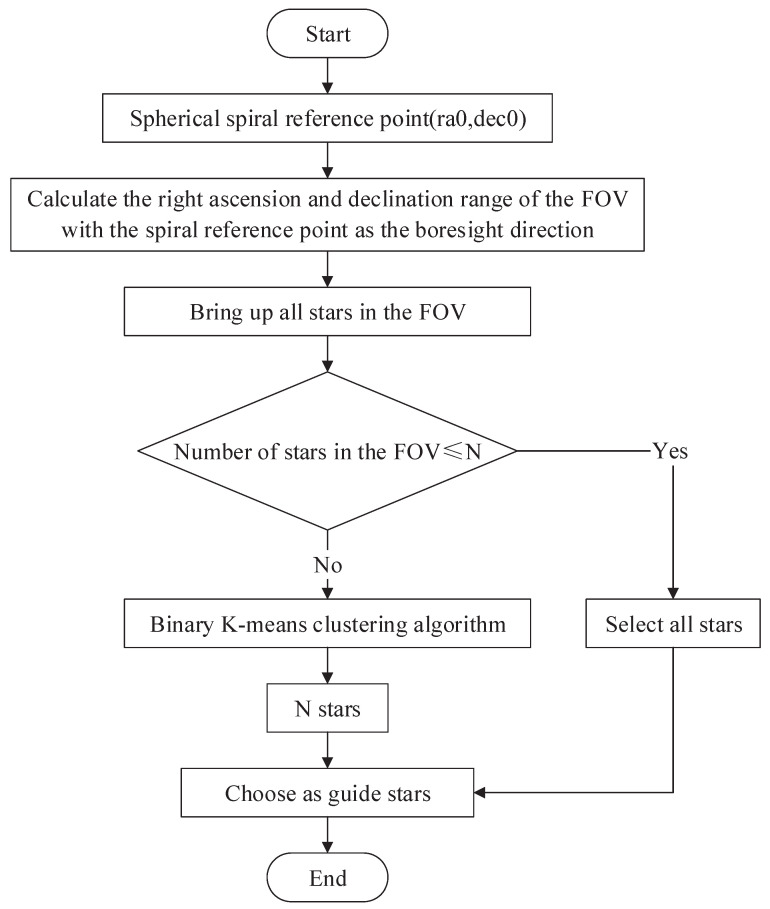
Flow chart of the SSBK.

**Figure 6 sensors-22-04996-f006:**
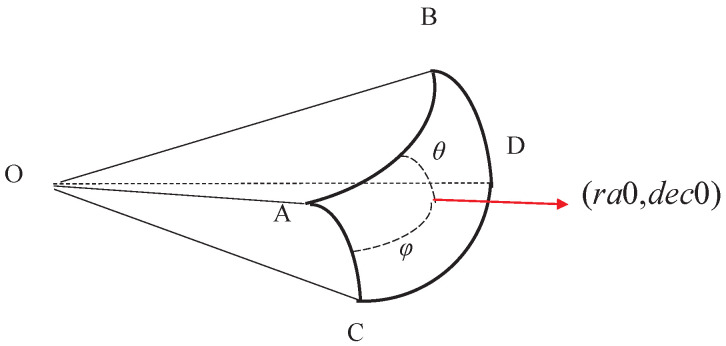
Schematic diagram of FOV corresponding to right ascension and declination.

**Figure 7 sensors-22-04996-f007:**
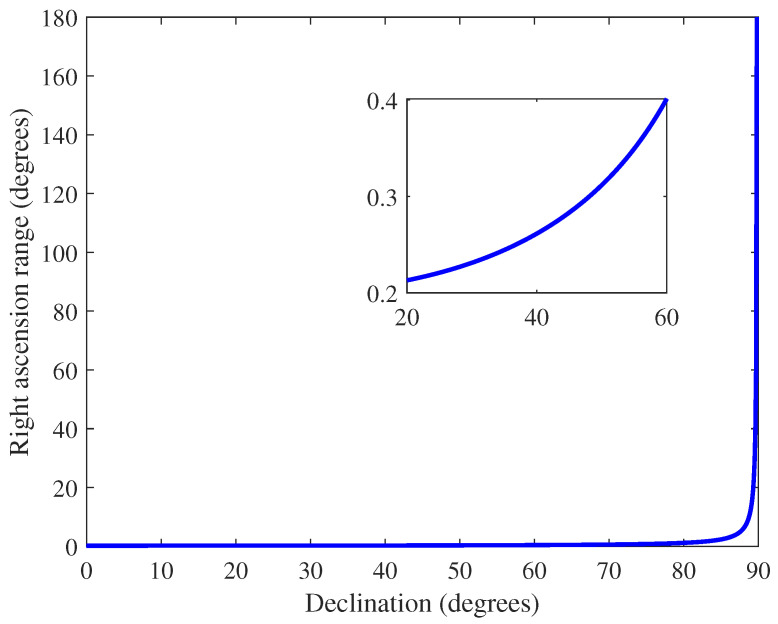
The curve of the range of right ascension corresponding to the field of view as declination changes (northern celestial sphere, FOV: 0.04 deg${}^{2}$).

**Figure 8 sensors-22-04996-f008:**
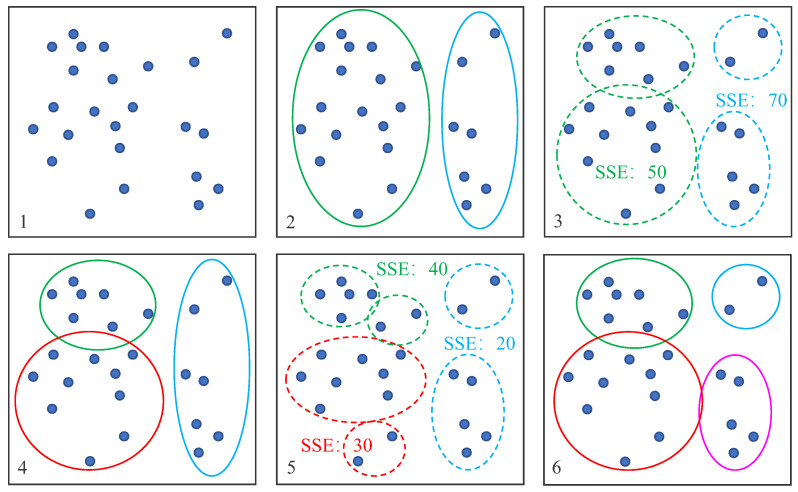
Algorithm schematic diagram. (**1**) Stars in the field of view; (**2**) Two clusters after binary K-means clustering; (**3**) Calculate the SSE after the two clusters are divided; (**4**) Three clusters after binary K-means clustering; (**5**) Calculate the SSE after the three clusters are divided; (**6**) Four clusters after binary K-means clustering.

**Figure 9 sensors-22-04996-f009:**
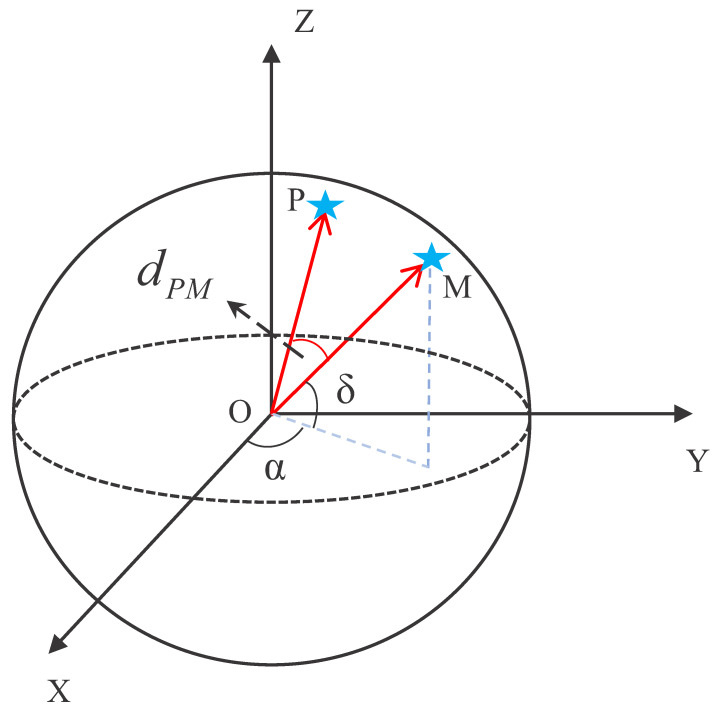
Schematic diagram of the angular distance.

**Figure 10 sensors-22-04996-f010:**
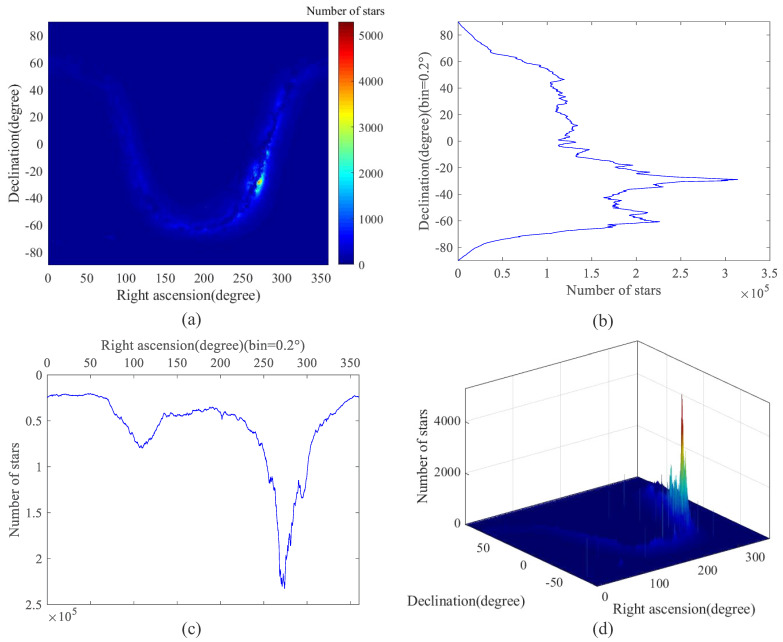
Star position distribution diagram of the candidate star catalog. (**a**) The distribution density of the candidate star catalog (bins = [1800,900]); (**b**) The statistics of the NOS distributed in the direction of declination; (**c**) The statistics of the NOS distributed in the direction of right ascension; (**d**) The three–dimensional distribution density of the candidate star catalog (bins = [1800,900]).

**Figure 11 sensors-22-04996-f011:**
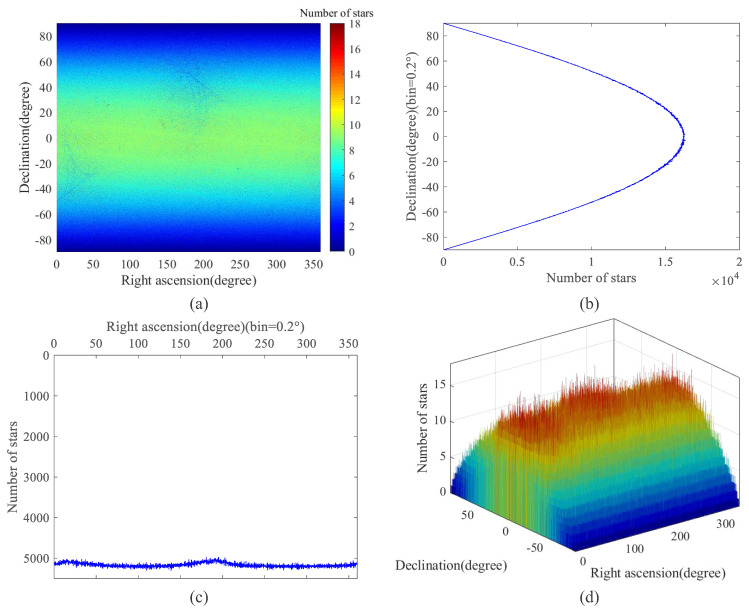
Guide star position distribution diagram of the guide star catalog. (**a**) The distribution density of the guide star catalog (bins = [1800,900]); (**b**) The statistics of the NOS distributed in the direction of declination; (**c**) The statistics of the NOS distributed in the direction of right ascension; (**d**) The three–dimensional distribution density of the guide star catalog (bins = [1800,900]).

**Figure 12 sensors-22-04996-f012:**
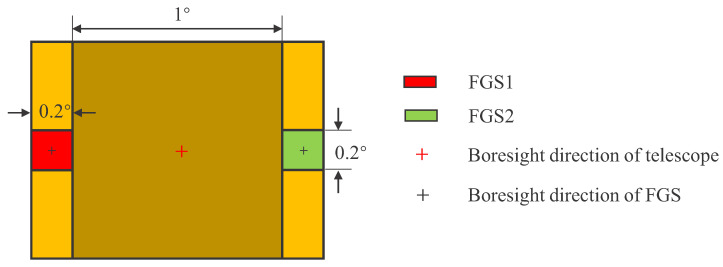
Diagram of position distribution of FGS.

**Table 1 sensors-22-04996-t001:** Basic characteristics of commonly used star catalogs.

Star Catalogs	Publication Year	Total Number of Stars	Epoch	Position Accuracy/Mas	Limit Magnitude/Mag	Average Star Density per Square Degree
SAO	1989	258,997	J2000.0	1000	9	6.278
Hipparcos	1997	118,218	J2000.0	1	12	2.866
Tycho-2	2000	2,539,913	J2000.0	7	11	61.569
GSC2.3	2006	945,592,683	J2000.0	200	20	22,921.813
UCAC4	2012	113,780,993	J2000.0	15–20 (M = 10–14) 70 (M = 16)	16	2758.107
GaiaDR2	2018	1,692,919,135	J2015.5	0.02–0.04 (M < 15) 0.1 (M = 17) 0.7 (M = 20)	21	41,037.518

**Table 2 sensors-22-04996-t002:** The simulation results of a single FOV tested in 10,000 random boresight direction.

Catalog	Method	Total Guide Stars	Number of Stars in FOV				Probability of Stars Appearing in FOV					Local Uniformity	Global Uniformity
			Min	Max	Mean	Std	<3	<5	5∼15	>15	>20		
Candidate Star Catalog	-	104,350,738	2	6104	101.89	179.10	0.0001	0.0009	0.0966	0.9025	0.8170	199.38	0.16
Guide	MFM	8,612,088	0	115	8.52	9.17	0.1931	0.4038	0.4516	0.1446	0.0913	8.76	0.11
Star	SSRP	9,279,614	1	18	8.98	2.28	0.0023	0.0023	0.9743	0.0034	0	2.23	7.82 × 10^−6^
Catalog	SSBK	9,298,862	2	18	9.01	1.69	0.0008	0.0057	0.9939	0.0004	0	1.61	1.91 × 10^−5^

**Table 3 sensors-22-04996-t003:** The simulation results of FGS1 tested in 10,000 random boresight direction.

Catalog	Method	Total Guide Stars	Number of Stars in FOV				Probability of Stars Appearing in FOV				
			Min	Max	Mean	Std	<3	<5	5∼15	>15	>20
Candidate Star Catalog	-	104,350,738	2	3740	101.19	170.44	0.0002	0.0007	0.0993	0.9000	0.8174
Guide	MFM	8,612,088	0	139	8.55	9.25	0.1932	0.4037	0.4527	0.1436	0.0903
Star	SSRP	9,279,614	1	20	8.98	2.28	0.0012	0.0215	0.9750	0.0035	0
Catalog	SSBK	9,298,862	2	18	9.0	1.67	0.0007	0.0052	0.9941	0.0007	0

**Table 4 sensors-22-04996-t004:** The simulation results of FGS2 tested in 10,000 random boresight direction.

Catalog	Method	Total Guide Stars	Number of Stars in FOV				Probability of Stars Appearing in FOV				
			Min	Max	Mean	Std	<3	<5	5∼15	>15	>20
Candidate Star Catalog	-	104,350,738	2	5031	101.15	174.04	0.0001	0.0001	0.1007	0.8983	0.8167
Guide	MFM	8,612,088	0	103	8.56	9.27	0.1931	0.4106	0.4438	0.1456	0.0937
Star	SSRP	9,279,614	1	19	9.0	2.28	0.0015	0.0203	0.9756	0.0041	0
Guide	SSBK	9,298,862	2	19	9.03	1.65	0.0006	0.0037	0.9959	0.0004	0

## Data Availability

Not applicable.

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
