# Peer review of "Construction of the Guide Star Catalog for Double Fine Guidance Sensors Based on SSBK Clustering"

_sensors, 2022, doi:10.3390/s22134996_

Round 1

Reviewer 1 Report

The authors presented the manuscript «Construction of the Guide Star Catalog for Double Fine Guidance Sensors Based on SSBK Clustering». The authors propose an original approach based on the spherical spiral binary K-means clustering algorithm that can be applied for determination of the speed of star identification and the accuracy of attitude determination of the space telescope. The abstract is written in sufficient volume and corresponds to the problem posed by the authors in the study. The approach used is described correctly. In general, the manuscript is well written. However, we can specify small comments.

-          Firstly, the authors often use abbreviations in the manuscript, which makes it difficult to be understood clearly, and i often has to refer to the meanings of the abbreviations.

-          The results of the simulation performed on a limited sample (Tables 2-4) are interesting. I would like to recommend the authors to expand the conclusions by including a detailed discussion of the simulation results obtained in the conclusions. Authors may numerically indicate std in conclusions for  examples.

-          Why do you use threshold for the NOS in the FOV = 9?

-Figure 8, I recommend say about axes smth. In this figure you have presented 6 subplots. But there is no detailed discussion in manuscript about it. Please explain this figure clearly.

Figure 11 d) does not contain color bar. Please add it.

-          What limitations and disadvantages does your approach have?

Author Response

My co-authors and I would like to thank you for your time and effort in reviewing the manuscript, as well as for your valuable suggestions. For the response please see the attachment.

Reviewer 2 Report

The article needs the following changes:

1) What are the important results?

2) What is the novelty of the work and where does it go beyond previous efforts in the literature?

3) What does the current paper add to the subject area compared with other published studies?

4) I think the results of the paper pave way to new avenues that are fully awaited. Therefore, the future works should be added in the conclusion part.

5) The changes made based on the comments should be written in color.

After carrying out these changes, I recommend that the paper can be published.

Author Response

(The authors gave the same response as above.)

Reviewer 3 Report

The reviewer is a specialist in machine and deep learning, not in astronomy, so my remarks are mainly to this area.

In general it is a really geed work. Just some questions or remarks for improvement.

1. The choice of k-means as the used machine learning learning algorithm is not justified, why this and no other one, here a short list of possible clustering algorithms used in machine learning:

  • Affinity Propagation
  • Agglomerative Clustering
  • BIRCH
  • DBSCAN
  • K-Means
  • Mini-Batch K-Means
  • Mean Shift
  • OPTICS
  • Spectral Clustering
  • Mixture of Gaussians (see https://machinelearningmastery.com/clustering-algorithms-with-python/)
  •  
  • Another point is the question about disadvantages of k-means: some of them are mentioned in the paper. Another important disadvantage of k-means is the fact, that algorithm badly deals with noise and outliers. It would be good to exclude this also.

Author Response

(The authors gave the same response as above.)

Reviewer 4 Report

I have the following comments and recommendations to the paper:

— Flow charts like Figure 5 are created as a bitmap copy from another software tool. I recommend such figures to be redone a vector-based pictures.

— The same situation in graphs like Figure 7 - all the letters in Figs. are actually bitmaps, I recommend to redraw the pictures using a vector-base drawing tool.

— Equations are made quite OK. I only recommend the abbreviations (in subscripts) to be written with Roman letter, e.g., please use \mathrm{FGS} in equations (4), (5), etc.

— The references are quite novel, recent, referencing is OK. However, some more detailed info about the references used for creating the tables at the end of the paper. (In detail, which references contain exactly all the numbers used in the tables.)

— Please insert a reference to the Monte Carlo method mentioned in line 308.

Generally, the paper seems to me interesting to the readers, and I recommend accepting the paper after minor corrections. 

Author Response

(The authors gave the same response as above.)

Round 2

Reviewer 2 Report

Accept.